# An Analysis of DNA Sequence Polymorphism in the Swamp Buffalo Toll-like Receptor (*TLR2*) Gene

**DOI:** 10.3390/ani13122012

**Published:** 2023-06-16

**Authors:** Wootichai Kenchaiwong, Pongphol Pongthaisong, Srinuan Kananit, Monchai Duangjinda, Wuttigrai Boonkum

**Affiliations:** 1Network Center for Animal Breeding and Omics Research, Khon Kaen University, Khon Kaen 40002, Thailand; 2Small Ruminant Research Unit, Faculty of Veterinary Science, Mahasarakham University, Mahasarakham 44000, Thailand; 3Department of Animal Science, Faculty of Agriculture, Khon Kean University, Khon Kean 40002, Thailand

**Keywords:** Toll-like receptors, diversity, polymorphism, swamp buffalo

## Abstract

**Simple Summary:**

This study will assist in the conservation of genetic resources and breeding planning for the sustainable utilization of Thai swamp buffalo. It is essential to clearly understand the diversity of genetic resources for breeding planning and genetic conservation. High genetic variation has the potential to exist, and vice versa; the species may easily become extinct when low genetic variation exists. The study revealed that the indicators of the genetic diversity of the swamp buffalo population were also genetically diverse in the *TLR2* coding region, as assessed by nucleotide diversity, and that genetic diversity was high. This indicates that it is easy to select and improve the available genetic tools to improve health and production in swamp buffalo when the populations are diverse. The authors suggest that *TLR2* gene variations may have the potential for marker-based breeding programs aimed at breeding better disease resistance in swamp buffalo.

**Abstract:**

Toll-like receptors (TLRs) are transmembrane proteins important for directing immune responses. Their primary role is to recognize pathogens based on single-nucleotide polymorphism (SNP) characteristics. TLR2 is categorized as a pattern recognition receptor (PRR) that is important for the recognition of pathogens. Nucleotide variation in the coding region determines the conformation of the TLR protein, affecting its protein domain efficiency. This study aimed to identify SNPs in the coding region of *TLR2* to enhance available genetic tools for improving health and production in swamp buffalo. A total of 50 buffaloes were randomly sampled from the northeastern part of Thailand for genomic DNA extraction and sequencing. Nucleotide sequences were aligned and compared with cattle and river buffalo based on the database. The results showed, there were 29 SNP locations in swamp buffalo and 14 different locations in both cattle and buffaloes. Haplotype analysis revealed that 27 haplotypes occurred. Swamp buffalo were identified from 13 SNPs based on biallelic analysis, which found eight synonymous and five nonsynonymous SNPs. Nucleotide diversity (π) was 0.16, indicating genetic diversity. Genetic diversity (haplotype diversity; HD) was high at 0.99 ± 0.04. This indicates a high probability that the two sample haplotypes are different. The π and HD values are important indicators of the genetic diversity of the swamp buffalo population. In summary, the Thai swamp buffalo population detected a polymorphism of the coding region of the *TRL2* gene. Therefore, further, in-depth study of the relationship between these genes in the immune system and disease resistance should be recommended.

## 1. Introduction 

The difference in chromosome numbers between swamp buffalo and river buffalo is one reason we are interested in performing molecular studies on them. Swamp buffaloes have 48 chromosomes, and river buffaloes have 50 chromosomes. Genetically diverse populations are easy to select and improve through breeding. Improvement through breeding is essential to clearly understand the diversity of genetic resources to formulate breeding plans and for genetic conservation [1]. Generally, Thai swamp buffaloes are raised in rural areas of Thailand. Selective breeding by artificial insemination is widespread. However, the most common method involves mating within a small herd, so there is likely to be a risk of loss of genetic diversity [2]. In 2021, Thailand’s estimated number of buffaloes was 1.52 million, with 72% distributed in the country’s northeast. Surin Province, in the northeast, has the highest population of buffaloes, with approximately 13% of the total buffaloes in the region [3]. The genetic resources management of domestic animals, such as chickens, pigs, dairy animals, and horses, has been made by intensive selection leading to genetic uniformity and causing several breeds to be at risk of extinction [4]. Hence, the chance of accidental genetic drift and inbreeding is high, and the decline in heterozygosity can decrease genetic diversity [5]. In addition, selective breeding by artificial insemination technology is more widespread at the farm level, so both natural and artificial selection influences genetic diversity [6]. Polymorphisms of single nucleotides play an essential role in natural selection and are also used as genetic markers in artificial selection for disease resistance. Genetic markers can indicate whether the risk is higher or lower than in the general population [7].

Water buffaloes (*Bubalus bubalis*) can be divided into two subspecies, swamp buffaloes (*B*. *bubalis carabanesis*) and river buffaloes (*B. bubalis bubalis*), and these two species have a different number of chromosomes. Swamp buffaloes have 48 chromosomes, and river buffaloes have 50 chromosomes. The worldwide buffalo population reported by the FAO in 2021 was estimated at 203.9 million head, a 1.37% increase from 2020 [3]. In Thailand, there are 1.52 million heads, representing 0.75% of all buffaloes currently distributed around the world [8]. In Thailand, most buffalos are swamp buffalos (2*n* = 48). They have adapted well to rural conditions where crop residues or local feed resources such as rice straw are used as the main feed. Moreover, meat from buffalo is relatively high in protein. Consequently, increasing production efficiency might enhance food security [9]. The genetic resource management of domestic animals, such as chickens, pigs, dairy animals, and horses, has employed intensive selection, leading to genetic uniformity and causing several breeds to be at risk of extinction [4]. In addition, selective breeding by artificial insemination technology is more widespread at the farmer level, and both natural selection and thus artificial selection also influence genetic diversity [6].

Single-nucleotide polymorphisms (SNPs) can be used for diversity studies. Moreover, SNPs of functional genes can also be developed for markers in a selection approach. Disease resistance traits are one of many characteristics that must be genetically improved in livestock. Several reports have mentioned that polymorphisms of the Toll-like receptor (*TLR*) gene are related to disease resistance and susceptibility in livestock [7]. 

The Toll-like receptor (TLR) family is a group of receptors that play a role in innate immune activation. TLRs are linked together in forming heterodimeric complexes that induce lipid conjugates, i.e., TLR2 and TLR1 or TlR6 [10]. Variations between 86–100% of nucleotide sequences and amino acids have been identified in ruminant TRLs [11,12]. TLRs are transmembrane proteins important for directing immune responses, playing a role as one of the pattern recognition receptors [13]. Based on ligand–pathogen specificity, ten TLRs have been identified in humans and livestock [14]. The expression of TLRs has been observed in several cells, such as peripheral blood mononuclear cells (PMMCs), spleen, lung and liver tissues, kidney, and reproductive tissues [11,15]. TLRs share a modular structure consisting of leucine-rich repeat (LRR) ectodomains, a single transmembrane helix, and a cytoplasmic Toll/interleukin-1 receptor (TIR) domain.

Among the TLRs reported in mammals, *TLR2* is a gene that plays a significant role in pathogen recognition based on the characteristics of an extracellular N-terminal domain, which consists of 16 to 28 leucine-rich repeats (LRRs) and the characteristics of the intracellular C-terminal domain or TIR domain needed to activate the downstream signaling pathway [7,16]. A previous study identified 29 new SNPs and their polymorphisms associated with tuberculosis (TB) infection in water buffalo [7]. *TLR2* is classified as a pattern recognition receptor (PRR), which has been previously described to be involved in recognizing bacteria, including mycobacteria, which cause the pathogenesis of tuberculosis. In addition, SNPs of *TLR2* have previously been identified, which have also been detected in buffaloes associated with disease resistance. In *TLR2* exon 2 region CDs, a region of genes reported to be found, 76% of SNPs were in the LRR protein domain, the region responsible for ligand recognition (Alfano et al. [7]). *TLR2* in Egyptian river buffaloes identified 26 SNP loci (2 in the 5′ UTR region, 22 in the coding region, and 2 in the 3′ UTR region), which were reported by Mossallam et al. [17]. Eight SNPs in the coding region have been reported in six species of buffaloes in India [18], whereas relatively few *TLR2* nucleotide polymorphisms have been reported in swamp buffaloes. Therefore, the present study aimed to identify SNPs in the coding sequences (CDs) of *TLR2* to enhance available genetic tools for improving health and production in swamp buffalo. 

## 2. Materials and Methods

### 2.1. Ethics and Animals

A total of 50 buffalos reared by farmers in five provinces of northeast Thailand (Sakon Nakhon, Nakhon Phanom, Khon Kaen, Surin, and Mahasarakham provinces) (Figure 1) were randomly sampled for genomic DNA extraction and genotyping. Blood samples were collected from the jugular vein into sterile tubes for DNA analysis. This study was approved by the Institutional Animal Care and Use Committee of Khon Kaen University (No. IACUC-KKU-40/62).

### 2.2. DNA Extraction and PCR Protocol

Ten milliliters of blood were collected from each buffalo and inserted into a microtube containing 10% 0.5 M EDTA as an anticoagulant. DNA was then extracted by using guanidine HCl as an adaptation of the method of Goodwin et al. [19] as follows: The extraction buffer (20% SDS 70 µL 7.5 M Na-acetate (50 µL), 1% Proteinase K (25 µL), and 7.5 M guanidine HCl (625 µL)) was mixed by vortexing for 5–10 s in 30 µL of white blood cells and then incubated at 60 °C in a thermo box for 2–3 h. After centrifugation at 6440× *g* for 5 min, the supernatant was aspirated into a new 1.5 mL microtube. A total of 600 µL of absolute cold isopropanol was added, and then the tube was flipped up/down gently, causing precipitation of DNA, and then centrifuged at 6440× *g* for 5 min. The supernatant was discarded, and the pellet was washed 2× with 1000 µL 75% ethanol. The DNA pellet was air-dried at room temperature and dissolved in 30–50 µL of TE buffer. The solution was incubated in a thermos box to dissolve DNA at 37 °C for 1–2 h. One microliter was aspirated to check DNA quality by Nanodrop, and the DNA solution was maintained at −20 °C.

The forward primer (F: TTTGTAGGTCAAATCACTGGACA) and reverse primer (R: TCCTGGCCACCGACA) were designed using the riverine buffalo database previously reported by Alfano et al. [7]. The PCR product was amplified in a total reaction volume of 10 µL containing 1 µL genomic DNA, 1.5 µL 10× buffer with MgCl2, 1 µL dNTP, 1 µL of each primer, and 0.25 U Taq Polymerase. The PCR protocol consisted of initial denaturation at 95 °C for 5 min followed by 35 cycles at 95 °C for 45 s, annealing temperature at 64 °C for 30 s, at 72 °C for 60 s, and final elongation steps at 72 °C for 1.30 min. The 1131 bp PCR product was detected using 2% agarose gel electrophoresis and visualized under a gel documentary system.

### 2.3. Sequencing and Bioinformatics Analysis

Sixty microliters of the pure PCR product were used for Sanger and nucleotide sequencing by sequencing analysis software version 7.0 (SqeqA7). The process was performed by ATGC Co. Ltd., Thailand Science Park (TSP) (Pathum Thani, Thailand). The predicted gene sequence was aligned against NCBI reference mRNA sequences of *Bubalus bubalis* using BLAST tools (http://blast.ncbi.nlm.nih.gov/Blast.cgi; accessed on 10 May 2021). Additionally, the predicted peptide sequence was aligned against nonredundant protein sequences by BLASTx (nucleotide-protein BLAST) to verify homology with other species and identify the predicted gene. Multiple alignments based on nucleotide sequences were performed using ClustalW via BioEdit 7.2 software [20,21]. The coordinates of genes were aligned and compared with a GenBank reference accession No database: HM756161. Haplotype phasing method analysis was performed using DnaSPv6.12.03 software, it is a program designed to work with DNA sequence polymorphism data, it can use data sets with polymorphic positions (e.g., SNP haplotypes), and it is also suitable for data with haploid phase input [22].

### 2.4. Genetic Diversity and Molecular Diversity Indices

Haplotype and genetic diversity were calculated with ARLEQUIN version 3.5.1.2 [23]. Genetic distance and phylogenetic tree with haplotype frequency in each grouping region for the *TLR2* gene were analyzed and constructed by unweighted pair group method with arithmetic mean (UPGMA) on NTSYS version 2.10.

## 3. Results

The allele frequency of swamp buffalo ranged from 0.06 to 0.94. The *TLR2* (exon2) gene partial nucleotide sequence of approximately 1131 base pairs was amplified from swamp buffalo and compared with reported sequences of other species. The swamp buffalo sequence exhibited 98.37% homology with the riverine buffalo *TLR2* gene (GenBank acc no. HM756161.1), 99.00% homology with the goat *TLR2* gene (GenBank acc no. NM-001285603.1), and only 97.12% homology with the bovine *TLR2* sequence (GenBank acc no. EU413951.1). The swamp buffalo *TLR2* gene has been characterized and found to code for a 784 amino acid protein similar to that of riverine buffalo.

### 3.1. Characterization of the TLR2 Exon 2 Gene in Swamp Buffaloes

A comparison of nucleotide sequences revealed that there were 14 different base sequence points between reference buffaloes (riverine) and swamp buffaloes compared to cattle, and 13 amino acids were altered because of the nucleotide sequence changes. Genetic polymorphisms by mutation site of the *TLR2* gene compared with reference nucleotide sequences of *Bubalus bubalis* were found for 29 SNP locations (Figure 2, blue letters) and 14 different locations between cattle and both types of buffalo (Figure 2, red letters). Based on the sequencing data, the SNP position can be confirmed randomly by a restriction enzyme. The present study selected the AvaII enzyme, which can cut at 2 locations covering g.G478A, and the Hind*I*II enzyme, which can cut at 2 locations covering T785C (Figure 3).

### 3.2. Haplotype (Hap) Analysis

The haplotype analysis revealed that 27 haplotypes occurred based on 29 SNP locations of *TLR2* exon 2. The most frequent cases were 14% of Hap7 (TCACAGAGTAGTATTTAACGCTACGACGT), 12% of Hap5 (TCACAGAGTAGTATCTAACGCTATGAGGT), 8% of Hap1 (TCACAGAGCAGTATCTAACGCTATGAGGT), Hap4 (TCACAGAGCAGTATCTAACGCTATGAGAT) and Hap14 (TCACAGAGTAGTATCTAACGCTATGAGAT), and 6% of Hap8 (TCACAGAGCAGTATTTAACGCTACGACGT), and the remaining haplotypes were less than 5%. This study found no haplotype patterns that appeared in all groups. The NP group buffaloes did not share haplotypes with any other group. However, six haplotypes were shared between groups. Hap5 and Hap7 shared haplotypes between 3 buffalo groups (SN, KK, and NP), and Hap1 was shared between the MK, SN, and KK groups. In addition, a shared haplotype was found between the two buffalo groups: Hap14 between the KK and Np groups and Hap4 and Hap8 between the SN and NP groups.

### 3.3. Genetic Diversity

The heterozygosity values of a population can indicate its genetic diversity. The current study found that the total heterozygosity of the buffalo divided into the northeastern part of Thailand was 0.16 (Table 1). The buffaloes in the MK region had the highest average expected heterozygosity (HE) values (0.24 ± 0.27), and KK buffaloes had the lowest (0.07 ± 0.0.15). In addition, it was found that average heterozygosity at locus 15 was the highest (0.52 ± 0.10), indicating that the locus had high genetic diversity.

### 3.4. Molecular Diversity Indices

Molecular diversity indices are shown in Table 2. Nucleotide diversity (π) was found to be 0.16 ± 0.09, indicating genetic polymorphism. However, genetic diversity (haplotype diversity) was high at 0.99 ± 0.04. The π value was found to be 0.16 ± 0.09. The polymorphic site-specific heterozygosity (He) expectation was 0.39 ± 0.15. In the whole population, the markers were in Hardy-Weinberg disequilibrium (*p* > 0.05). The parameters of a neutrality test were investigated by the method of Tajima’s D [24] and Fu’s FS test [25]. A deviation from neutral equilibrium revealed Tajima’s D value of 0.23 (*p* > 0.05) and Fu’s FS of −2.52 (*p* > 0.05), indicating that the buffalo population had not deviated from neutral equilibrium.

### 3.5. Genetic Structure

The study of the genetic structure of the population is presented in Table 3. Genetic structural differences were found when analyzed from the total population; Fst = 0.239, and there was an intragroup variation of 77.46%. Analysis of genetic differences from the pairwise Fst values revealed that the buffalo population of the SR group was significantly different from that of other groups (Table 4). A phylogenetic tree was constructed using genetic distance based on haplotype frequency in each group region for the TLR2 gene analyzed using the UPGMA method. The results showed that buffaloes could be classified into two groups; each group did not depend on the proximity of provinces to each other (Figure 4).

## 4. Discussion

In a previous study, the % identity in the coding region of the *TLR2* gene between swamp and river buffaloes was 96.6%, cattle 92.7%, and the highest 97.9% in sheep [26], which is consistent within this study, the identity (%) of the *TLR2* gene between swamp buffaloes and goats was also high (99%). While the amino acid identity (%) between swamp buffaloes and river buffaloes was the highest at 98.9%, cattle at 97.1%, and goats at 96.9%, reported by Shi et al. [26]. Dubey et al. [27] reported that maximum homology (99%) of the 5′ upstream region of the buffalo *TLR* sequence with that of the goat was observed to be in concordance with previous reports on *TLR8*.

The swamp buffalo *TLR2* exon 2 sequence exhibited 98.30% homology with that of the riverine buffalo, initiated from the sequence 1171–3525 referenced by GenBank Acc. No HM756161. Genetic polymorphisms by mutation site of the *TLR2* gene were found for 29 SNP locations. The number of SNP locations (only in the part we studied) was more than that in the previous report in swamp buffaloes, whereas only ten mutations were found in riverine buffaloes [7]. Several SNPs can be confirmed with PCR-RFLP, including g.478G > A/AvaII and g.780T > C/HindIII. PCR-RFLP simplifies the use of marker genes for candidate gene marker studies.

Only one part was used to translate into the first 438 amino acids. This study is the first report finding 13 SNPs in *TLR2* (exon 2) in swamp buffalo. In this study, the positions were found to be different from those reported in the water buffalo of [7], which revealed 13 synonymous and five nonsynonymous positions in water buffaloes. However, our results found five SNP positions, which is consistent with the results of Alfano et al. [7], who reported that in riverine buffalo, 53C > T (rs1388116475:C > T), 108C > T (rs1388116476:C > T), 374T > C (rs1388116479:T > C), 519G > C (rs1388116482:G > C), and 1034A > G (rs13881 16475:A > G). The region responsible for ligand recognition is the protein domain (LRR); approximately 76.00% of SNPs were reported to be detected for *TLR2*, *TLR4*, and *TLR9* [7]. Synonymous mutations can change transcription processes by reducing the binding of RNA-binding proteins, while silent mutations can interfere with translating mRNA into protein. However, it may not be directly related to disease tolerance or susceptibility but may be related to other genes [7]. Only the part used to translate into the first 438 amino acids was used for haplotype reconstruction based on *TLR2* polymorphisms performed by DnaSPv6.12.03 software. This generated 27 possible haplotypes. We found six haplotypes with a frequency greater than 6%, which displayed a frequency of approximately 2% in the remaining haplotypes. Previously, the haplotype CTTACCAGCGGGCCAGTCCCCC was reported to be associated with disease resistance by Alfano et al. [7]. We found that the first ten nucleotides of the reference haplotype pattern were the exact coverage locations of the primer pair used in this study.

Expected heterozygosity (HE) is a general statistic used to estimate heterozygosity within populations, High HE levels also indicate high genetic diversity. The nucleotide diversity was found to be 0.16, indicating a high level of genetic diversity. As Nei [28] defined, the full range can be 0.008–0.01, the medium range is 0.005–0.007, and the low range is 0.001–0.004. Additionally, genetic diversity (haplotype diversity) values were high at 0.99, with reference values > 0.5 to 1.0 indicating high-diversity groups. On the other hand, a value from 0.0 to >0.5 is a low-diversity range [29]. The π and HD values are important in determining the genetic diversity of a population. The differences in π and HD values result from the influence of the mutation rate, the mixing of populations from different geographical locations, overlapping generations, natural selection, and genetic drift [30]. We report an average Tajima’D of 0.23 (−0.38 to 0.65), which is close to the value reported in native animals by Chen et al. [31] reported in the range of −1.44 to 2.92 in the *TLR* family gene (*TLR3* and *TLR8*) in five cattle breeds native to Yunnan. The average HD was 0.99 (range 1.0 to 0.93), with high diversity similarity as *TLR* families reported high HD values (0.88) in native cattle from Yunnan (Dengchuan and Dianzhong). Nucleotide variations show more potential utilization in genetic selection by use as marker genes. In previous studies, nucleotide variations in the bovine and buffalo *TLR2* gene were associated with somatic cell counts, mastitis, and tuberculosis [27,32,33,34].

The fixation index (Fst) has been estimated as a measure of genetic differentiation between populations, with a high value of Fst (close to 1) indicating high genetic differentiation between populations, the two populations do not share genetic diversification, or the populations are fixed. On the other hand, a low value of Fst (close to 0) indicates a complete distribution of the genetic material. Criteria for distinguishing the degree of differentiation of Fst values are small genetic difference (Fst < 0.05), moderate genetic difference (Fst = 0.05 to 0.15), great genetic difference (Fst = 0.15 to 0.25) and very great genetic difference (Fst > 0.25) reported by Hartl and Clark [35]. We found a high level of genetic diversity in Thai swamp buffaloes based on a relatively high mean Fst (0.23) using *TLR2* gene SNP data from 29 loci. Diversity declines have been reported in 17 populations of Turkey’s river buffaloes, with an estimated low Fst of 0.032 based on a study using 20 microsatellite markers [1]. An Fst of 0.13 has been reported in African buffaloes, a level of difference between buffalo populations that need to be controlled [36].

Molecular variance analysis performed by AMOVA revealed that individual diversity within the population represents approximately 77% of the total of all genetic variations and approximately 22% of all genetic variations clarified by differences among the population. The proportion of molecular variance in swamp buffaloes with individuals within the population in our report was less than that reported in Turkey’s river buffaloes, which was approximately 88% based on microsatellites [1].

The pairwise Fst was shown the genetic distances. The Fst values of pairwise comparisons among the 5 Thai swamp buffalo ranged from 0.000 to 0.657, which is a large extent compared to the previous report by Üral et al. [1] in the Turkish buffalo (ranged from 0.00 to 0.0866). The construction of a phylogenetic tree is a standard method to represent the relatedness of genes. It is generally claimed that genetic data DNA sequencing for analysis provides a true relationship between taxa. The classification of swamp buffalo groups by DNA sequencing of *TLR2* genes can be divided into two groups. Genetic affinity did not arise from neighboring provinces. Genetic diversity in members of the *TLR* family needs to be shown with clinical data to increase the likelihood of selective utilization from endemic animal genetic sources.

## 5. Conclusions

This study revealed the accelerating conservation of genetic resources if overall population diversity is reduced, and genetic management is implemented. This finding will help lead to further studies on the relationship between these genes and other important traits in swamp buffalo, such as immune response, disease susceptibility, or resistance.

## Figures and Tables

**Figure 1 animals-13-02012-f001:**
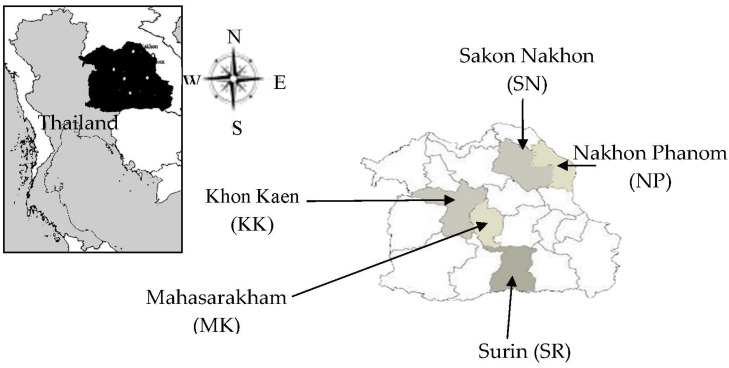
The location of blood and DNA sampling in Thailand.

**Figure 2 animals-13-02012-f002:**
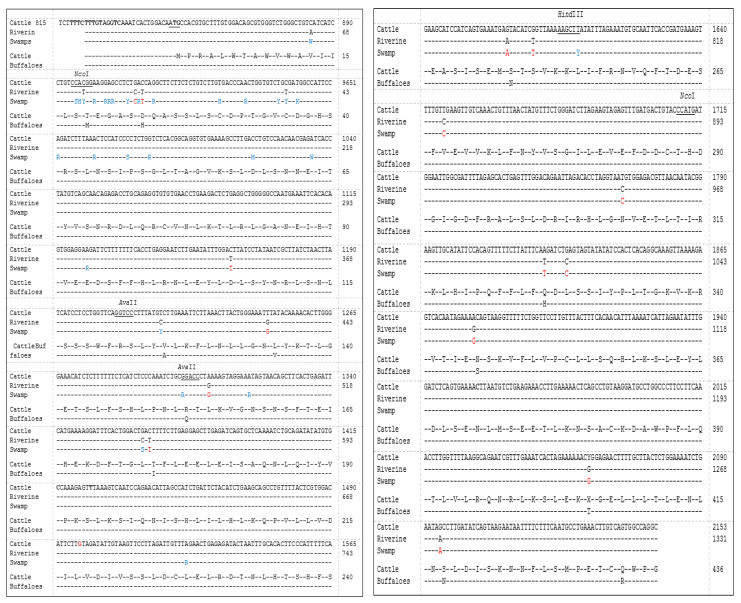
Multiple alignment nucleotide sequences (top 3 lines) of the *Bubalus bubalis TRL2* (exon2) gene by using Clustral Omerga version 1.2.2 software for identification between swamp buffaloes and river buffaloes (GenBank accession No. HM756161) and cattle (GeneBank accession No. EU746461), the blue letters are the SNPs of swamp buffaloes and the red letters are the same nucleotides in the two groups of buffalo but different from the cattle; protein alignment (last 2 lines) between cattle (GenBank accession No. ACH92790) and river buffaloes (GenBank accession No. ADO51627), different letters are different amino acids.

**Figure 3 animals-13-02012-f003:**
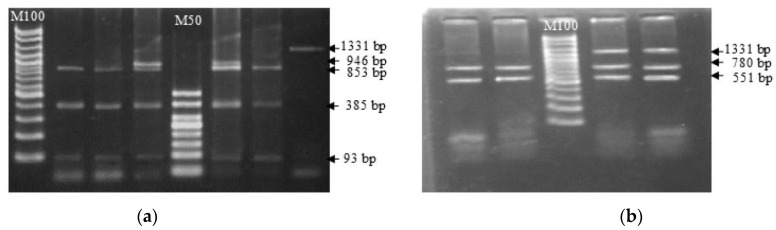
Identification of SNPs in g. G478A and g. T785 of the swamp buffalo *TLR2* exon 2 gene by PCR-RFLP with *Ava*II (**a**) and *Hind*III (**b**) restriction enzymes.

**Figure 4 animals-13-02012-f004:**
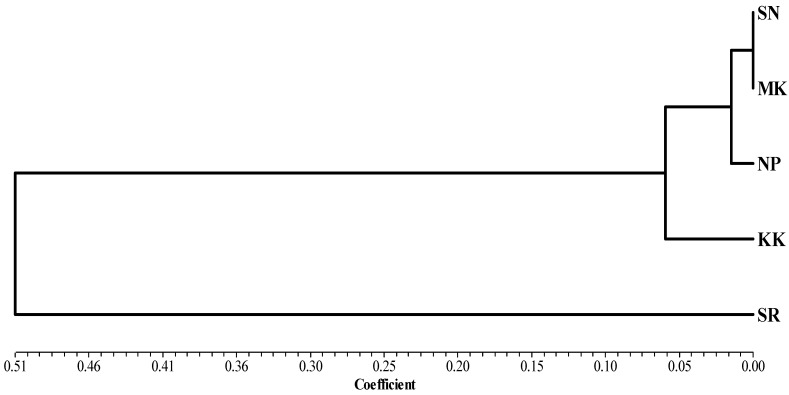
Phylogenetic analysis by UPGMA based on the genetic distance of haplotype frequency in each grouping region of the *TLR2* gene using NTSYS version 2.10.

**Table 1 animals-13-02012-t001:** The expected heterozygosity of the Thai swamp buffaloes from five provinces.

SNPs	Expected Heterozygosity						H_E_	H_O_	Fixation Indices (Fst)
	SN (*n* = 10)	MK (*n* = 6)	KK (*n* = 10)	NP (*n* = 18)	SR (*n* = 6)	Mean ± SD			
A39T	0	0.53	0	0.21	0	0.16 ± 0.26	0.31	0.36	0.60
C51G	0	0.33	0	0.21	0	0.12 ± 0.18	0.12	0.12	0.00
A52C/T	0.47	0.60	0	0	0	0.23 ± 0.33	0.19	0.20	0.18
T53C	0	0	0	0	0.60	0.12 ± 0.27	0.12	0.15	0.60 *
A56G	0	0.33	0	0	0	0.08 ± 0.18	0.04	0.04	0.09
A59G	0	0.33	0	0	0.40	0.16 ± 0.22	0.08	0.09	0.09
A60G	0	0.33	0	0	0	0.08 ± 0.18	0.04	0.04	0.09
A61G	0	0.33	0	0	0	0.08 ± 0.18	0.04	0.04	0.09
C65T	0.47	0.60	0.36	0.53	0	0.39 ± 0.24	0.50	0.52	0.17
A68G	0	0	0	0.11	0	0.02 ± 0.05	0.04	0.04	0.00
G72A	0.20	0	0	0	0	0.04 ± 0.09	0.04	0.04	0.00
T90A/C	0	0	0	0	0.40	0.08 ± 0.18	0.19	0.26	0.85 *
A97G	0	0.33	0	0	0	0.08 ± 0.18	0.04	0.04	0.09
T106C	0	0	0	0	0.40	0.08 ± 0.18	0.04	0.05	0.14
T108C	0.53	0.53	0.36	0.50	0.60	0.52 ± 0.10	0.49	0.50	0.00 *
T111G	0	0	0	0	0.40	0.08 ± 0.18	0.15	0.21	0.81 *
A121G	0.20	0	0	0	0	0.04 ± 0.09	0.04	0.04	0.00
A131G	0	0.33	0	0	0	0.08 ± 0.18	0.04	0.04	0.09
C140G	0.20	0	0	0	0	0.04 ± 0.09	0.04	0.04	0.00
G146A	0	0	0	0	0.40	0.08 ± 0.18	0.15	0.21	0.81 *
C174A	0	0	0	0	0.40	0.08 ± 0.18	0.04	0.05	0.14
A190T	0	0	0	0	0.40	0.08 ± 0.18	0.04	0.05	0.14
A279G	0.20	0	0	0	0	0.04 ± 0.09	0.04	0.04	0.00
C374T	0.56	0.53	0.36	0.52	0	0.41 ± 0.25	0.50	0.52	0.14
G455A	0.47	0	0	0.11	0	0.12 ± 0.20	0.15	0.15	0.12
A473G	0	0	0	0	0.60	0.12 ± 0.27	0.08	0.10	0.38 *
C519G	0.53	0.53	0.36	0.52	0	0.40 ± 0.24	0.50	0.51	0.14
A681R	0.53	0.60	0.47	0.50	0.40	0.50 ± 0.07	0.47	0.47	0.08 *
T762C	0.36	0	0	0.21	0	0.11 ± 0.16	0.15	0.15	0.00
average	0.16 ± 0.21	0.24 ± 0.21	0.07 ± 0.21	0.12 ± 0.21	0.17 ± 0.21	0.15 ± 0.21	0.16 ± 0.16	0.17 ± 0.17	
						χ^2^ = 0.10, (χ^2^_0.05,28_ = 41.34)			

SN = Sakon Nakhon; MK = Mahasarakham; KK = Khon Kaen; NP = Nakhon phanom; SR = Surin; H_E_ = expected heterozygosity; H_o_ = observed heterozygosity. * SNPs position starting 1 from ATG start codon.

**Table 2 animals-13-02012-t002:** The estimated parameters of genetic diversity in the Thai swamp buffalo TLR2 exon 2 gene from five locations.

Statistics/Locations	SN	MK	KK	NP	SR	Average ± SD
Diversity indices						
No. of substitutions	12	15	5	10	11	10.4
No. private subst. sites	4	5	0	1	8	3.6
Genetic diversity	1.00 ± 0.04	1.00 ± 0.12	1.00 ± 0.04	0.93 ± 0.04	1.00 ± 0.13	0.99 ± 0.04
Nucleotide diversity	0.16 ± 0.09	0.24 ± 0.16	0.06 ± 0.09	0.12 ± 0.07	0.17 ± 0.12	0.16 ± 0.09
H_E_	0.39 ± 0.15	0.51 ± 0.11	0.38 ± 0.05	0.34 ± 0.19	0.45 ± 0.09	0.39 ± 0.15
π	4.71 ± 2.84	6.27 ± 4.69	1.89 ± 1.32	3.43 ± 2.06	5.00 ± 3.41	4.26 ± 1.66
Neutrality test						
Tajima’s D test						
Tajima’s D	0.49	0.14	0.28	0.65	−0.38	0.23 ± 0.39
Tajima’s D *p* value	0.70	0.59	0.66	0.75	0.44	0.63 ± 0.11
Fu’s FS test						
Exp. no. of alleles	5.72	4.46	3.94	6.73	3.73	4.82
FS	−6.17	−1.73	0.47	−3.81	−1.35	−2.52 ± 2.54
FS *p* value	0.001	0.090	0.599	0.011	0.122	0.165

SN = Sakon Nakhon; MK = Mahasarakham; KK = Khon Kaen; NP = Nakhon phanom; SR = Surin; π is the average number of pairwise differences; nucleotide diversity is the average over loci; H_E_ is expected heterozygosity.

**Table 3 animals-13-02012-t003:** Analysis of molecular variance analysis (AMOVA) for computing conventional F-statistics based on haplotype frequencies.

Source of Variation	df	SS	Variance Components	Percentage of Variation	Fixation Indices	*p* Value
Among populations	4	28.41	0.56 Va	22.543	F_ST_: 0.239	*p* < 0.001
Within populations	44	84.53	1.92 Vb	77.46		
Total	48	112.94	2.48			

**Table 4 animals-13-02012-t004:** Population pairwise (FSTs) between buffaloes from five locations by computing conventional F-statistics from haplotype frequencies.

Locations	SN	MK	KK	NP	SR
SN	0.000				
MK	0.000	0.000			
KK	0.127	0.031	0.000		
NP	0.0.03	0.000	0.024	0.000	
SR	0.459 *	0.394 *	0.657 *	0.523 *	0.000

SN = Sakon Nakhon; MK = Mahasarakham; KK = Khon Kaen; NP = Nakhon phanom; SR = Surin; * Significance (Matrix of significant Fst; *p* values significance level = 0.05).

## Data Availability

The data presented in this study are available on request from the Network Center for Animal Breeding and Omics Research, Faculty of Agriculture, Khon Kaen University, Thailand.

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
