# Peer review of "An Analysis of DNA Sequence Polymorphism in the Swamp Buffalo Toll-like Receptor (TLR2) Gene"

_animals, 2023, doi:10.3390/ani13122012_

Round 1

Reviewer 1 Report

The main conclution of  concept of "indicators of genetic diversity  pf the swamp buffalo population  were also geneticaly diverse in TLR 2 coding region  as assesed by the nucleotide diversity was high " can be used as selection tools  in other words  markers for favourable genotypes.

So this would be nice findings .so far selection criteria for water buffalo mainly targeted  and based on phenotypic values which requires same time interval to obtain .So in this way authors focussed diminish the generation interval and so increas the response per year in selection studiees .I recommend make and some explanation  about the meaning of f statistics such as fixation indices of FST etc ,meaning of lack of heterozygosity minimum and maximum values of  Fst( (fixation of sub population and total population)etc.Although sampled population size seems small(50 Buffaloes for 4 different locality) limits the resuts of  findings(statistics9 of samples to expand population parameteres  but as a preliminary findins the results hopefull for appling breeding studies 

Author Response

Dear Reviewer,

Response to Reviewer 1 Comments

Point 1: The main conclusion of concept of "indicators of genetic diversity pf the swamp buffalo population were also genetically diverse in TLR 2 coding region as assessed by the nucleotide diversity was high " can be used as selection tools in other words markers for favorable genotypes. So, this would be nice findings.so far selection criteria for water buffalo mainly targeted and based on phenotypic values which requires same time interval to obtain. So, in this way authors focused diminish the generation interval and so increase the response per year in selection studies.

Response 1: We thank you for spending your valuable time reviewing our manuscript and for understanding the importance of this research. We are trying to find suitable ways to use the genetic resources of native buffaloes for sustainable genetic development. Certainly, indicators of swamp buffalo population genetic diversity will help plan conservation and breeding of swamp buffaloes. Establishing conservation guidelines if the population is less diverse. Selection strategies for genetic improvement are considered sustainable production efficiency improvements. We chose to study polymorphisms based on the TLR2 gene, which plays an important role in disease tolerance. Most often found in river buffaloes. While rarely reported in swamp buffaloes, this work is among the first to report on swamp buffaloes.

Point 2: I recommend make and some explanation about the meaning of f statistics such as fixation indices of FST etc, meaning of lack of heterozygosity minimum and maximum values of  Fst (fixation of sub population and total population) etc.

Response 2: We added more information about the meaning, the explanation must be made in the full text as your suggestion. please see the revised manuscript.

Lines 339-340: “Expected heterozygosity (HE) is a general statistic used to estimate heterozygosity within populations, High HE levels also indicate high genetic diversity.”

Lines 352-359: “The fixation index (Fst) has been estimated as a measure of genetic differentiation between populations, with a high value of Fst (close to 1) indicating high genetic differentiation between populations, the two populations do not share genetic diversification, or the populations are fixed. On the other hand, a low value of Fst (close to 0) indicates a complete distribution of the genetic material. Criteria for distinguishing the degree of differentiation of Fst values that are small genetic different (Fst<0.05), moderate genetic different (Fst=0.05 to 0.15), the great genetic different (Fst=0.15 to 0.25) and very great genetic different (Fst>0.25) reported by Hartl and Clark (1997).” 

Lines 371-374: “The pairwise FST were shown the genetic distances. The FST values of pairwise comparisons among the 5 Thai swamp buffalo were ranged from 0.000 to 0.657, which is a large extent compared to the previous report by Ural et al. (2021) in the Turkish buffalo (ranged from 0.00 to 0.0866).”

Reference papers

Hartl, D.L. and Clark, G.C. (1997) Principles of Population Genetics. Sinauer Associates, Sunderland.

Ünal EÖ, Işık R, Åžen A, Geyik KuÅŸ E, Soysal MÄ°. Evaluation of Genetic Diversity and Structure of Turkish Water Buffalo Population by Using 20 Microsatellite Markers. Animals (Basel). 2021;11(4):1067.

Point 3: Although sampled population size seems small (50 Buffaloes for 4 different locality) limits the results of findings (statistics 9 of samples to expand population parameters but as a preliminary finding the results hopefully for applying breeding studies.

Response 3: We agree with you about the sample size used for genetic diversity research. Obviously, the larger the sample size, the more reliable it is. However, when comparing the statistical numbers of this study with those of previous studies, the values were in a similar range. We also check the accuracy of the data and numbers from the analysis results, such as PCR-RFLP sequencing results were validated, including SNP confirmation against existing databases. Therefore, we are confident that even though the sample size is small, the statistical analysis results can be used as academic references.

Best Regards

Wuttigrai Boonkum

Reviewer 2 Report

The study is unique in terms of examining the Toll-like Receptor (TLR2) Gene region polymorphism related to swamp buffaloes in Thailand. However, considering that a small number of individuals were studied in the study, it is thought that the observed number of SNPs is quite high. How do you interpret this situation? In a few sentences, the reasons for this should be explained.

 To strengthen the originality of the study, it is suggested that the following corrections be made in the article.

-          The contribution of the study of the related gene region (TLR2) polymorphism for swamp buffaloes should be expressed in a few sentences in the introduction.

-           It is reported that 29 different SNPs were observed in the study. It would be appropriate to display the relevant SNPs on a reference sequence.

-          - It is reported that 50 swamp buffaloes were studied in the study and these samples were taken from 5 different provinces. It is thought that the sample numbers collected according to the provinces should be indicated in Table 1.

-          At the same time, SNP should be written instead of “locus” in Table 1.

-          In the material method part of the study: “Genetic distance with haplotype frequency in each grouping region for the TLR2 gene was analyzed by unweighted pair group method with arithmetic mean (UPGMA) on NTSYS version 2.10. A phylogenetic tree was constructed with genetic distance by the neighbor-joining (NJ) method using NTSYS software.” an expression is included. However, in the results section, it is seen that the phylogenetic tree is given according to the UPGMA method. Which one is correct UPGMA or NJT???

-          Why was the goat TLR2 gene region, which was reported to be 99% homology to Figure 2, not added?

-          Have DNA sequences containing all haplotypes been uploaded to the database ((NCBI) ? If uploaded, gene bank numbers should be specified in the article. If not, it is recommended to upload the data to the gene bank.

Author Response

Dear Reviewer,

Response to Reviewer 2 Comments

Point 1: The study is unique in terms of examining the Toll-like Receptor (TLR2) Gene region polymorphism related to swamp buffaloes in Thailand. However, considering that a small number of individuals were studied in the study, it is thought that the observed number of SNPs is quite high. How do you interpret this situation? In a few sentences, the reasons for this should be explained.

Response 1: About the observed number of SNPs is quite high, the number of 29 SNPs of TLR2 was not quite high, a number similar to that reported by Alfano et al. (2014), who identified 29 SNPs in river buffalo and within the range of the number of SNPs in cattle (3 to 43 SNPs), reviewed by Novak (2014).

Reference papers:

  1. Alfano, F.; Peletto, S.; Lucibelli, M.G.; Borriello, G.; Urciuolo, G.; Maniaci, M.G.; Desiato, R.; Tarantino, M.; Barone, A.; Pasquali, P.; Acutis P.L.; Galiero, G. Identification of single nucleotide polymorphisms in Toll-like receptor candidate genes associated with tuberculosis infection in water buffalo (Bubalus bubalis). BMC Genet.2014, 139, 1-8.
  2. Karel Novak. 2014. Functional polymorphisms in Toll-like receptor genes for innate immunity in farm animals Veterinary Immunology and Immunopathology 157 (2014) 1– 11.

Point 2:  The contribution of the study of the related gene region (TLR2) polymorphism for swamp buffaloes should be expressed in a few sentences in the introduction.

Response 2. We added sentences in the introduction part. See lines 106-111 “TLR2 in Egyptian river buffaloes identified 26 SNP loci (2 in the 5` UTR region, 22 in the coding region, and 2 in the 3` UTR region), which were reported by Mossallam et al. (2022). Eight SNPs in the coding region have been reported in six species of buffaloes in India (Banerjee et al.,2012), whereas relatively few TLR2 nucleotide polymorphisms have been reported in swamp buffaloes.”

Point 3: It is reported that 29 different SNPs were observed in the study. It would be appropriate to display the relevant SNPs on a reference sequence.

Response 3: We have shown the appropriate representation of the 29 SNPs loci, indicating their loci and nucleotides. Please see Table 1.

Point 4: It is reported that 50 swamp buffaloes were studied in the study and these samples were taken from 5 different provinces. It is thought that the sample numbers collected according to the provinces should be indicated in Table 1. At the same time, SNP should be written instead of “locus” in Table 1.

Response 4:  We have edited. Please see Table1.

Point 5: In the material method part of the study: “Genetic distance with haplotype frequency in each grouping region for the TLR2 gene was analyzed by unweighted pair group method with arithmetic mean (UPGMA) on NTSYS version 2.10. A phylogenetic tree was constructed with genetic distance by the neighbor-joining (NJ) method using NTSYS software.” an expression is included. However, in the results section, it is seen that the phylogenetic tree is given according to the UPGMA method. Which one is correct UPGMA or NJT???

Response 5: We have edited and rewritten it more clear in materials and methods part. Please see lines 167-169. “Genetic distance and phylogenetic tree with haplotype frequency in each grouping region for the TLR2 gene was analyzed constructed by unweighted pair group method with arithmetic mean (UPGMA) on NTSYS version 2.10.”

Point 6:  Why was the goat TLR2 gene region, which was reported to be 99% homology to Figure 2, not added?

Response 6: We have verified that we have identified TLR genes that are predisposed to homology with goats, such as the study by Shi et al. (20) and Dubey et al. (24). However, considering the physiological similarities between swamp buffaloes, river buffaloes and goats, we chose to compare only the two groups. Please see lines 304-309.

“In a previous study, the % identity in the coding region of the TLR2 gene between swamp and river buffaloes was 96.6%, cattle 92.7%, and the highest 97.9% in sheep (Shi et al. 2020), which is consistent with In this study, the identity (%) of the TLR2 gene between swamp buffaloes and goats was also high (99%). While the amino acid identity (%) between swamp buffaloes and river buffaloes was the highest at 98.9%, cattle at 97.1%, and goats at 96.9%, reported by Shi et al. 2020.”

Point 7: Have DNA sequences containing all haplotypes been uploaded to the database ((NCBI) ? If uploaded, gene bank numbers should be specified in the article. If not, it is recommended to upload the data to the gene bank.

Response7: Thank you for your suggestion. The consensus sequence has already been prepared. We plan to upload data to the gene bank.

Best Regards,

Wuttigrai Boonkum

Reviewer 3 Report

The manuscript provides some information regarding the SNPs in TLR2 genes in Thai buffaloes. The sample size is small and no association analyses were performed. In addition, much information is not clear such as how did the author define haplotypes and did the authors sequence the whole coding regions or just exon 4. I do not see the manuscript is enough for publication in Animals as a full paper.

Line 25: Might shortly introduce the roles of TLR2 as the study is focusing on TLR2 only.

Line 25: Why only the coding region

Which types of sequence, and aligned and compared with which?

How many SNPs.

Line 29-30 “The results ….” This sentence does not have meaning,

Line 34: What is the meaning of HD in genetic diversity

Line 36: What does it mean position?

Did the authors test the association with the disease,, if not how it can be used for the selection

Table 1: Put the number of animals for each province, what does it mean locus here? The table should be more in the results section. The authors should provide the positions and the alleles of the SNPs.

What are the differences between the buffaloes in different provinces?

Line 149; Why only exon 4

Figure 4: What are the bootstrap values?

How did the authors define haplotypes?

Line 262-282: it should not be in the discussion, it is not related to any results from the current study

Line 283-289: Similarly, this paragraph should be in the introduction.

Line 324: Pi

Line 328: Which TLR genes

Line 331: change TLR to TLR2

Author Response

Dear Reviewer,

Response to Reviewer 3 Comments

Point 1: The manuscript provides some information regarding the SNPs in TLR2 genes in Thai buffaloes. The sample size is small and no association analyses were performed. In addition, much information is not clear such as how did the author define haplotypes and did the authors sequence the whole coding regions or just exon 2. I do not see the manuscript is enough for publication in Animals as a full paper.

Response 1: We thank you for spending your valuable time reviewing our manuscript and we understand your concerns.

Obviously, the larger the sample size, the more reliable it is. However, when comparing the statistical numbers of this study with those of previous studies (reference in MS), the values were in a similar range. We also check the accuracy of the data and numbers from the analysis results, such as PCR-RFLP sequencing results, including SNP confirmation against existing databases. Therefore, we are confident that even though the sample size is small, the statistical analysis results can be used as academic references.

Moreover, about define haplotypes of this study, the identification of haplotypes based on the positional differences of SNP1-SNP29 of the TLR2 genes after Multiple Sequence Alignment (MSA) analysis of the DnsV6 software package.

For sequence the whole coding regions or just exon 2, we studied in Exon 2 because in TLR2 exon 2 region CDs, a region of genes reported to be found, 76% of SNPs were in the LRR protein domain, the region responsible for ligand recognition.

Finally, from we mentioned above, we believe this study will lead to several benefits, including:

  1. It gives us the knowledge of a gene that is important for conservation and genetic utilization in swamp buffalo, which is reported in a small number compared to river buffaloes.
  2. Helping breeders to plan the selection and breeding of swamp buffalo correctly and appropriately.
  3. For farmers will have good breeding buffalo to generate income.

Point 2: Line 25: Might shortly introduce the roles of TLR2 as the study is focusing on TLR2 only.

Response 2: We added more information the role of TLR2 gene in revised MS. Please see lines 24-26. “TLR2 is categorized as a pattern recognition receptor (PRR) that is important for the recognition of pathogens. Nucleotide variation in the coding region determines the conformation of the TLR protein, affecting its protein domain efficiency.”

Point 3: Line 25: Why only the coding region. Which types of sequence, and aligned and compared with which?

Response 3: Nucleotide sequences was aligned and compared with cattle and river buffalo based on database. Lines 29-30.

How many SNPs.

“There were 29 SNP locations in swamp buffalo” Lines 30-31.

Point 4: Line 29-30 “The results ….” This sentence does not have meaning,

Response 4: We already deleted.

Point 5: Line 34: What is the meaning of HD in genetic diversity.

Response 5: We have explained this further in line 34.

“Genetic diversity (HD) was high at 0.99±0.04. This indicates a high probability that the two sample haplotypes are different.”

Point 6: Line 36: What does it mean position?

Response 6: We have added more position data. See line 37.

Point 7: Line 36: Did the authors test the association with the disease, if not how it can be used for the selection.

Response 7: We rewritten the conclusion of this study to avoid the overstated because we do not yet study about the association with the disease. Please see line 36-39.

Point 8: Table 1: Put the number of animals for each province, what does it mean locus here? The table should be more in the results section. The authors should provide the positions and the alleles of the SNPs.

Response 8: We have added information based on your suggestions. See Table 1.

Point 9: What are the differences between the buffaloes in different provinces?

Response 9: We used the criterion to select the study area from the number of buffaloes that is higher than in other provinces. While the phenotypic characteristics of Thai swamp buffaloes in all provinces are similar. However, this study expects that different raising conditions, areas, and feed resources may affect to genetic variation.

Point 10: Line 149; Why only exon 2

Response 10: We have provided more details in the introduction section. Please look at the line 100-106

“TLR2is classified as a pattern recognition receptor (PRR), which has been previously described to be involved in recognizing bacteria, including mycobacteria, which cause the pathogenesis of tuberculosis. In addition, SNPs of TLR2 have previously been identified, which have also been detected in buffaloes associated with disease resistance. In TLR2 exon 2 region CDs, a region of genes reported to be found, 76% of SNPs were in the LRR protein domain, the region responsible for ligand recognition (Alfano et al (2014)”.

Point 11: What are the bootstrap values?

Response 11:  We used distance data for each group in the study. The bootstrap value requires an additional program, but due to the limitation of the program we use can't display the value so we only give the coefficient.

Point 12: How did the authors define haplotypes?

Response 12: Identification of haplotypes based on the positional differences of SNP1-SNP29 of the TLR2 genes after Multiple Sequence Alignment (MSA) analysis of the DnsV6 software package.

Point 13: Line 262-282: it should not be in the discussion, it is not related to any results from the current study

Response 13: We moved to introduction part. Please see lines 43-63

Point 14: Line 283-289: Similarly, this paragraph should be in the introduction.

Response 14:  We moved to introduction part. Please see lines 84-87

Point 15: Line 324: Pi

Response 15: We have changed the word from Pi to π. See lines 346, 347

Point 16: Line 328: Which TLR genes

Response 16: We have changed the word from TLR genes to TLR2 gene. See lines 351-352.

Point 17: Line 331: change TLR to TLR2

Response 17: We have changed the word from TLR genes to TLR2 gene. See lines 361-362.

Best Regards

Wuttigrai Boonkum

Reviewer 4 Report

The authors aimed to identify SNPs in the coding region of Toll-like receptor (TLR) 2. TLR2 is a membrane receptor that is expressed on the cell surface. It recognizes PAMPs (e.g. bacterial, fungal and viral components) and transmits appropriate signals to the immune system for activation.

In the submitted study by Kenchaiwong and coworkers, a total of 50 swamp buffaloes were collected from the northeastern part of Thailand to extract and sequence genomic DNA. TLR2 SNPs were used for diversity studies to measure haplotypes and genetic diversity. The results showed the distribution of SNPs in the coding region of the TLR2 gene. The authors were able to identify 13 SPNs based on biallelic analysis, which revealed eight synonymous and five non-synonymous SPNs.

The authors conclude that SNP positions in the TLR2 gene could be used as markers for potential disease resistance selection in swamp buffalo.

Given the complexities of pathogen-host interactions and the immune system, this bold statement is clearly overstated and should be removed from the manuscript. The data show only a detectable genetic variation of TLR2 sequences; whether this is actually of immune-functional significance will have to be shown in appropriate future studies, which are likely to be experimentally very complex.

Although I think that the rather limited work of the study is purely descriptive and of interest only to a very small group of scientists, I have no real scientific objections to the submitted manuscript (in terms of scope and journal ambition). Therefore, in my view, it can be published as is by MDPI Animals.

Author Response

Dear Reviewer,

Response to Reviewer 4 Comments

The authors aimed to identify SNPs in the coding region of Toll-like receptor (TLR) 2. TLR2 is a membrane receptor that is expressed on the cell surface. It recognizes PAMPs (e.g. bacterial, fungal and viral components) and transmits appropriate signals to the immune system for activation.

In the submitted study by Kenchaiwong and coworkers, a total of 50 swamp buffaloes were collected from the northeastern part of Thailand to extract and sequence genomic DNA. TLR2 SNPs were used for diversity studies to measure haplotypes and genetic diversity. The results showed the distribution of SNPs in the coding region of the TLR2 gene. The authors were able to identify 13 SPNs based on biallelic analysis, which revealed eight synonymous and five non-synonymous SNPs.

Point 1: The authors conclude that SNP positions in “the TLR2 gene could be used as markers for potential disease resistance selection in swamp buffalo.”

Given the complexities of pathogen-host interactions and the immune system, this bold statement is clearly overstated and should be removed from the manuscript. The data show only a detectable genetic variation of TLR2 sequences; whether this is actually of immune-functional significance will have to be shown in appropriate future studies, which are likely to be experimentally very complex.

Response 1: We have removed and revised the conclusions of this study. Please see lines 36-39 “In summary, the Thai swamp buffalo population detected a polymorphism of the TRL2 gene. Therefore, in further, in-depth study of the relationship of these genes in immune system and disease resistance should be recommended.” And lines 384-386 “This finding will help lead to further studies on the relationship between these genes and other important traits in swamp buffalo, such as immune response, disease susceptibility or resistance.”

Point 2: Although I think that the rather limited work of the study is purely descriptive and of interest only to a very small group of scientists, I have no real scientific objections to the submitted manuscript (in terms of scope and journal ambition). Therefore, in my view, it can be published as is by MDPI Animals.

Response 2: We thank you for spending your valuable time reviewing our manuscript and we understand your concerns.

Obviously, the larger the sample size, the more reliable it is. However, when comparing the statistical numbers of this study with those of previous studies (reference in MS), the values were in a similar range. We also check the accuracy of the data and numbers from the analysis results, such as PCR-RFLP sequencing results, including SNP confirmation against existing databases. Therefore, we are confident that even though the sample size is small, the statistical analysis results can be used as academic references.

Finally, from we mentioned above, we believe this study will lead to several benefits, including:

  1. It gives us the knowledge of a gene that is important for conservation and genetic utilization in swamp buffalo, which is reported in a small number compared to river buffaloes.
  2. Helping breeders to plan the selection and breeding of swamp buffalo correctly and appropriately.
  3. For farmers will have good breeding buffalo to generate income.

Best Regards,

Wuttgrai Boonkum

Round 2

Reviewer 3 Report

Thank the authors for responding.  Most of my comments have been addressed, however, I still have one major concern about power tests for sample size/analyses in point 1.  

Point 1: As the authors mentioned about the confidence in the power of statistical analyses and sample size, could the authors provide the statistical tests for it? What did the authors mean by “the values were in a similar range” could the authors provide more supporting references for the statement?

 Point 12: Could the authors provide the information on how the software defines haplotypes, why did the authors choose this method and software?

Author Response

Dear Reviewer,

We completed answer the questions please see the detail below.

Response to Reviewer 3 Comments

Thank the authors for responding.  Most of my comments have been addressed, however, I still have one major concern about power tests for sample size/analyses in point 1.  

Point 1: As the authors mentioned about the confidence in the power of statistical analyses and sample size, could the authors provide the statistical tests for it? What did the authors mean by “the values were in a similar range” could the authors provide more supporting references for the statement?

In order to complete answer your questions we would like to divide the response into 3 answers as follows:

Point 1.1: could the authors provide the statistical tests for it?

Response 1.1:

A larger sample size generally leads to increased power of test. Therefore, we started by calculating a sample size in this study, which we used the sample size calculation for unknown population following equation:

Where,  is Z-value at 95% confidence level is equal 1.96, is genetic diversity standard deviation from several studies = 0.04 and  is estimation error value equal 1.0% (or in the other words, equal 99% accuracy of statistical estimation) = 0.01.

Although the appropriate sample size from calculation should be about 61-62 head. However, in our study, we therefore considered a total of 50 samples (10 each from the 5 study areas) due to the limitations in blood sample data collection, meanwhile when compared to previous studies on the same gene in buffalo populations from many countries for example le Roex et al. (2017) used 43 African buffalo blood samples originated in the Kruger National Park, South Africa to study the Toll-like receptor (TLR) diversity influences mycobacterial growth in African buffalo, Mossallam et al. (2022) used 40 Egyptian river buffaloes to study polymorphism evaluation of TLR2 gene associated with endometritis infection in buffalo reared in Egypt, Tantia et al. (2012) used 24 animals from 6 diverse Indian water buffalo breeds (Murrah, Bhadawari, Tarai, Pandharpuri, Marathwada and Mehsana) to study phylogenetic and sequence analysis of toll like receptor genes (TLR-2 and TLR-4) in buffaloes, and Tanamati et al. (2019) used 24 Murrah buffaloes from a farm located in Registro, São Paulo State, Brazil to study Differential expression of immune response genes associated with subclinical mastitis in dairy buffaloes. For that reason, we considered that the number of our animals (50 sample size) used in this study was sufficient to study TLR2 gene polymorphisms.    

Supporting references:

[36] le Roex, L.; Jolles, A.; Beechler, B.; van Helden, P.; Hoal, E. Toll-like receptor (TLR) diversity influences mycobacterial growth in African buffalo. Tuberculosis. 2017, 104, 87–94

[17] Mossallam, A.A.A.; Osman, N.M.; Othman, O.E.; Mahfouz, E.R. Polymorphism evaluation of TLR2 gene associated with endometritis infection in buffalo reared in Egypt. Biotechnol. J. Int. 2022, 26, 45–55.

Tantia, M.S.; Mishra, B.; Banerjee, P.; Joshi, J.; Upasna, S.; Vijh, R.K. Phylogenetic and sequence analysis of toll like receptor genes (TLR-2 and TLR-4) in buffaloes. Indian J. Anim. Sci. 2012, 82, 875–878.

Tanamati, F.; Stafuzza, N.B.; Gimenez, D.F.J.; Stella, A.A.S.; Santos, D.J.A.; Ferro, M.I.T.; Albuquerque, L.G.; Gasparino, E.; Tonhati, H. Differential expression of immune response genes associated with subclinical mastitis in dairy buffaloes. Animal. 2019, 13, 1651–1657.

Point 1.2: What did the authors mean by “the values were in a similar range”

Response 1.2: The explanation for “the values were in a similar range” in this study is as follows.

“the values were in a similar range” we would like to describe the number of SPNs we found close to the study of the TLR2 gene in the 40 Egyptian buffaloes identified, 26 SNPs loci (2 in the 5` UTR region, 22 in the coding region, and 2 in the 3` UTR region), which were reported by Mossallam et al. (2022).” Please see in lines 106-108.

“Our results found five SNP positions, which is consistent with the results of Alfano et al. [6], who reported that in riverine buffalo, 53C > T (rs1388116475:C > T), 108C > T (rs1388116476:C > T), 374T > C (rs1388116479:T > C), 519G > C (rs1388116482:G > C), and 1034A > G (rs13881 16475:A > G).” Please see in lines 323-327.

“It also describes, the % identity in the coding region of the TLR2 gene between swamp and river buffaloes was 96.6%, cattle 92.7%, and the highest 97.9% in sheep (Shi et al. 2020), which is consistent with in this study, the identity (%) of the TLR2 gene between swamp buffaloes and goats was also high (99%).” Please see in lines 304-307.

“We report an average Tajima'D of 0.23 (-0.38 to 0.65), which is close to the value reported in native animals by Chen et al. (2020) reported in the range of -1.44 to 2.92 in the TLR family gene (TLR3 and TLR8) in five cattle breeds native to Yunnan. The average Hd was 0.99 (range 1.0 to 0.93), with high diversity similarity as TLR families reported high Hd values (0.88) in native cattle from Yunnan (Dengchuan and Dianzhong).” Please see in lines 350-354.

Point 1.3: could the authors provide more supporting references for the statement?

Response 1.3: Supporting references:

[17] Mossallam, A.A.A.; Osman, N.M.; Othman, O.E.; Mahfouz, E.R. Polymorphism evaluation of TLR2 gene associated with endometritis infection in buffalo reared in Egypt. Biotechnol. J. Int. 2022, 26, 45–55.

[26] Shi, W.; Mei, X.; Kang, Y.; Elsheikha, H.M.; Hu, C.; Wang, Y.; Lu, K.; Zhang, Y.; Sheng, Z.; Wang, D.; Zhu, X.; Huang, W. De novo characterization of the genetic polymorphism and transcript abundance of Toll-like receptors (TLRs) in tissues of swamp buffaloes (Bubalus bubalis) from Guangxi, China. Research square. 2022, 1–33.

[31] Chen, Y.; Yang, Y.; Li, C.; Li, R.; Xiao, H.; Chen, S. Genetic diversity of TLR3 and TLR8 genes among five Chinese native cattle breeds from southwest China. Livest. Sci. 2020, 232, 103895.

[7] Alfano, F.; Peletto, S.; Lucibelli, M.G.; Borriello, G.; Urciuolo, G.; Maniaci, M.G.; Desiato, R.; Tarantino, M.; Barone, A.; Pasquali, P.; Acutis P.L.; Galiero, G. Identification of single nucleotide polymorphisms in Toll-like receptor candidate genes associated with tuberculosis infection in water buffalo (Bubalus bubalis). BMC Genet. 2014, 139, 1-8.

Point 12: Could the authors provide the information on how the software defines haplotypes, why did the authors choose this method and software?

Response 12: 

DNA Sequence Polymorphism version 6 (DnaSP6 software) is a widely used software designed for the analysis of DNA sequence data. DnaSP6 is the latest version of this software. It offers a range of functionalities that make it valuable for researchers studying molecular evolution, population genetics, genetic diversity, and phylogenetics. The software defines haplotypes by refers to the process of statistical estimation of haplotypes from genotype data (also known as "phasing"). Therefore, phased haplotype data is required for use in the program. Phased haplotype was the most accurate method as well as it was the first method to utilize ideas from coalescent theory concerning the joint distribution of haplotypes. This method used a Gibbs sampling approach in which each individual haplotypes were updated conditional upon the current estimates of haplotypes from all other samples.

There are reasons why many researchers and us choose to use DnaSP6 software in this study.

  1. DnaSP6 software allows researchers to estimate various measures of genetic diversity, such as nucleotide diversity, haplotype diversity, and Tajima's D. These measures provide insights into the levels and patterns of genetic variation within and between populations.
  2. DnaSP6 software includes methods for detecting signatures of natural selection, such as the McDonald-Kreitman test and the Hudson-Kreitman-Aguadé test. These tests compare synonymous and non-synonymous substitutions to assess whether natural selection has acted on specific genes or genomic regions.
  3. DnaSP6 software allows researchers to reconstruct phylogenetic trees based on DNA sequence data. It supports various methods, including neighbor-joining and maximum likelihood, enabling the investigation of evolutionary relationships and the reconstruction of ancestral sequences.
  4. The software provides graphical outputs, such as haplotype networks, frequency spectra, and diversity plots, which help researchers visualize and interpret their results effectively.

In conclusion, DnaSP6 software is a versatile tool that facilitates the analysis and interpretation of DNA sequence data in evolutionary and population genetics studies. It enables researchers to explore genetic diversity, detect selection, infer population structure, and reconstruct evolutionary relationships, contributing to our understanding of molecular evolution and genetic processes.

We added the sentence for more information “Haplotype phasing method analysis was performed using DnaSP6 software, it is a program designed to work with DNA sequence polymorphism data, it can use data sets with polymorphic positions (e.g., SNP haplotypes), and it is also suitable for data with haploid phase input [22]. Please see in lines 162-165.

Best Regards,

Wuttigrai Boonkum
